

# Fluctuations in coral reef fish densities after environmental disturbances on the northern Great Barrier Reef

Zegni Triki and Redouan Bshary

Institute of Biology, University of Neuchâtel, Neuchâtel, NE, Switzerland

## ABSTRACT

Global warming is predicted to increase the frequency and or severity of many disturbances including cyclones, storms, and prolonged heatwaves. The coral reef at Lizard Island, part of the Great Barrier Reef, has been recently exposed to a sequence of severe tropical cyclones (i.e., Ita in 2014 and Nathan in 2015) and a coral bleaching in the year 2016. Reef fishes are an essential part of the coral reef ecosystem, and their abundance is thus a good marker to estimate the magnitude of such disturbances. Here, we examined whether the recent disturbances at Lizard Island had an impact on the coral reef fish communities. To do this, we examined fish survey data collected before and after the disturbances for potential changes in total fish density post-disturbance. Also, by sorting fish species into 11 functional groups based on their trophic level (i.e., diet), we further explored the density changes within each functional group. Our findings showed an overall decline of 68% in fish density post-disturbance, with a significant density decrease in nine of 11 trophic groups. These nine groups were: browsers, corallivores, detritivores, excavator/scrapers, grazers, macro-invertivores, pisci-invertivores, planktivores, and spongivores. The piscivores, on the other hand, were the only "winners," wherein their density showed an increase post-disturbance. These changes within functional groups might have a further impact on the trophodynamics of the food web. In summary, our findings provide evidence that the fish assemblage on the reefs around Lizard Island was considerably affected by extreme weather events, leading to changes in the functional composition of the reef fish assemblage.

# INTRODUCTION

The recently observed increase in frequency and magnitude of extreme weather events is attributed to anthropogenic global warming (*Cai et al., 2014*; *Cheal et al., 2017*; *Hughes et al., 2018*). Such extreme events are a great threat to coral reefs worldwide (*Hughes et al., 2017*). Coral reefs are one of the world's most diverse ecosystems, with fish as an essential component. Losing live corals can thus have severe impacts on the diversity and stability of this ecosystem (*Bellwood et al., 2006*; *Pratchett et al., 2008*, *2011*; *Munday et al., 2008*). For instance, one of the threats of extreme weather events to

Corresponding author
Zegni Triki, zegni.triki@unine.ch

coral reefs is the prolonged El Niño cycles and the resulting coral bleaching. El Niño is a naturally occurring climatic event that brings warm water toward the Indo-Pacific. A recent prolonged El Niño event led to an increase in seawater temperatures (*Cai et al., 2014*; *Hoegh-Guldberg & Ridgway, 2016*). In these conditions, overstressed coral tissues expel their intracellular symbionts "zooxanthella" (i.e., symbionts from which corals gain their different pigmentations) which causes bleaching. The resulting bleached corals may die if they do not re-establish the symbiotic relationship with the zooxanthella within a range of 6 months post-bleaching (*Diaz-Pulido & McCook, 2002*). In addition to the threat of coral bleaching, cyclones can also be destructive due to the formation of strong waves that can damage exposed coral reef fields (*Cheal et al., 2017*). Both cyclones and coral bleaching can thus result in environmental degradation and habitat loss (*Pizarro et al., 2017*; *Hughes et al., 2017*).

Using fish assemblages, diversity, and abundance, researchers can evaluate the biological integrity and quality of a given habitat (*Karr, 1981*; *Ganasan & Hughes, 1998*). Several studies, for instance, showed that fish abundance could be negatively affected by environmental disturbances due to climate change, either directly through abiotic factors such as temperature and ocean acidification (*Ferrari et al., 2011*; *Browman, 2016*), or indirectly through habitat loss (*Munday et al., 2008*). Thus, changes in fish abundance should provide reliable information on habitat quality.

Habitat degradation is known to have a negative impact on overall fish density (*Munday, 2004*; *Wilson et al., 2008b*, *2010*). *Bellwood et al. (2004)* argue that further insights can be gained from analyzing fish functional groups but only in addition to knowing the cause and extent of the habitat degradation. Therefore, exploring potential changes at the level of fish groups that share the same function (i.e., functional group) might yield additional information about the mechanism and effect of the impact. For instance, three main functional groups displaying herbivore dietary traits (i.e., corallivores, excavator/scrapers, and grazers) can play an important role in coral reef recovery. The functional role of these three herbivores is complementary, and together their presence on the reef can play a role in its resistance to disturbances (*Bellwood et al., 2004*). In addition to the densities of herbivorous fishes, other factors also play a major role in coral reef resistance and recovery, such as the complexity of coral structure and water depth (*Graham et al., 2015*).

A suitable location to explore potential changes in fish abundance and functional groups after environmental disturbances is Lizard Island (*Pizarro et al., 2017*; *Emslie, Cheal & Logan, 2017*; *Triki et al., 2018*). The island is located in the northern Great Barrier Reef (GBR), Australia, within a marine reserve. The island was impacted by a sequence of extreme weather events three years in a row: In April 2014, Cyclone Ita hit Lizard Island (*Pizarro et al., 2017*), reaching an intensity of category 5 on the Australian scale (*Puotinen et al., 2016*). In April 2015, the island was again exposed to another severe cyclone, Cyclone Nathan, a severe category 4 cyclone (*Pizarro et al., 2017*). And finally, in February/March of 2016, the GBR was affected by a massive coral bleaching event, resulting in more than 60% bleached coral cover (*Hughes et al., 2017*).

In this study, we asked to what extent fish communities would change as a function of environmental disturbances at Lizard Island. To do so, we compared fish densities before and after disturbances both overall and by functional group. We expected to find a decline in fish species that rely directly or indirectly on live corals for their diet (*Wilson et al., 2006*). In contrast, due to the colonization of dead corals by microalgae (*Cheal et al., 2010*) we expected an increase in the abundance of various herbivorous fish species specialized on such algae (*Randall, 1961*).

## METHODS

### Field site and fish census

The study was conducted on the reef around Lizard Island, GBR, Australia (14.6682°S, 145.4604°E). The study was carried out at two locations: Mermaid Cove and Northern Horseshoe reefs. Mermaid Cove forms a continuous fringing reef of approximatively 35,000 m² (i.e., estimated from maps: https://www.freemaptools.com/area-calculator.htm), with a depth range from one to seven m. The reef is located in a small bay on the northern side of Lizard Island. The other location, Northern Horseshoe reef, is also a continuous reef, consisting of a coral garden of approximately 17,000 m², with a depth range from one to four m. The reef is located on the western side of the island (see Fig. 1). After the 2014 and 2015 cyclones, the reef at Mermaid Cove was heavily damaged. Northern Horseshoe reef, however, had been protected from these two cyclones due to its location within the lagoon (*Pizarro et al., 2017*; Lizard Island Research Station Directors, Dr. Anne Hoggett and Dr. Lyle Vail in 2018, personal communication). The coral bleaching event in 2016 affected all the reefs around Lizard Island including our two study sites.

We used underwater visual fish census methods based on earlier studies by *Wismer et al. (2014)* and *Triki et al. (2018)*. Within each location, the observer swam ten replicates of a 30 m transect line on the reef flat. Due to the different shape of the reef at the two locations, the transect line was placed parallel to the reef crest at Mermaid Cove, whereas at Northern Horseshoe it was placed parallel to the shoreline (Following methods in *Wismer et al., 2014*). Along the 30 m transect line the observer first recorded the number of all large visible fish (i.e., species with body total length TL >10 cm) on a five m wide area, then the number of small visible fish (i.e., species with body TL ≤10 cm) on a one m wide area. Each of the ten transect replicates, within each location, were sampled at least 10 m apart from each other to minimize possible resampling of the same individuals. Only adult fish were surveyed, and their species was identified. Overall, there were 163 species identified in our survey (Table S1). All fish counts (i.e., large and small fish) were scaled per 150 m² to facilitate further statistical analyses.

The fish surveys from the two study locations were collected at the same time of day in a similar way between June and August within each year of data collection. Data were collected at Mermaid Cove in 2011 (in *Wismer et al., 2014*), 2016 (in *Triki et al., 2018*), and 2017. At Northern Horseshoe, the fish census was conducted in 2014, 2016 (in *Triki et al., 2018*) and 2017. We labeled the data collected in 2011 (i.e., from Mermaid

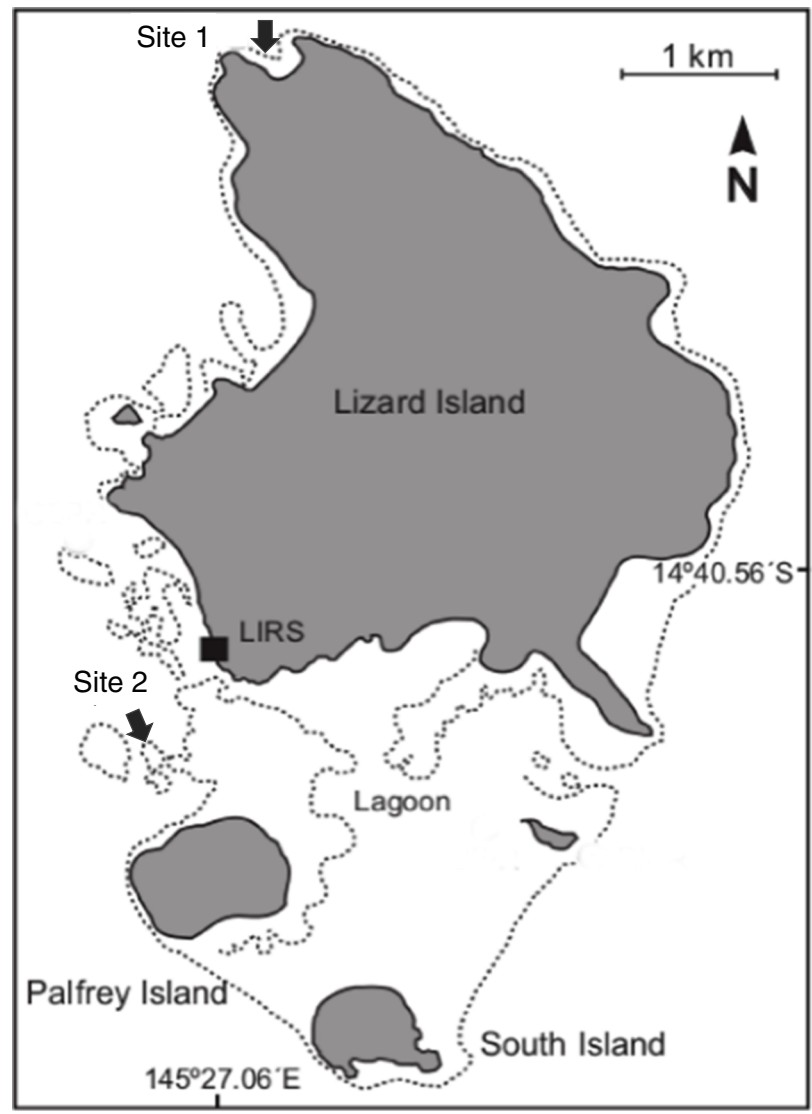

**Figure 1 Lizard Island group map.** The map is showing the two study sites: Mermaid Cove reef as Site 1 and Northern Horseshoe reef as Site 2. Modified from *Triki et al. (2018)*, *Global Change Biology* (© 2017 John Wiley & Sons Ltd).          

Cove) and 2014 (i.e., from Northern Horseshoe), as fish survey before disturbances. Subsequently, the fish surveys in 2016 and 2017 were labeled as data after the disturbances (as in *Triki et al., 2018*).

Fish species were then categorized into functional groups based on the species' trophic level (*Butterfield & Suding, 2013*; *Brandl et al., 2016*). We sorted fish species into 11 trophic-functional groups (Table 1). The categorization into dietary functional groups followed methods in studies by *Wernberg et al. (2013)* and *MacNeil et al. (2015)* (see Table S1). For the few species for which trophic level was missing from these studies, we completed information from the FishBase (*Froese & Pauly, 2016*).

**Table 1 Dietary functional trait used in sorting fish species into trophic-functional groups.**

| Trophic-functional group | Diet | Example |
|---|---|---|
| Browsers | Macro-algae | *Naso unicornis* |
| Corallivores | Corals | *Chaetodon aureofasciatus* |
| Detritivores | Dead organic material "detritus" | *Ctenochaetus striatus* |
| Excavators/scrapers | Remove reef substrate while looking for living material | *Chlorurus spilurus* |
| Grazers | Fast-growing macro-algae "turf algae" | *Siganus doliatus* |
| Macro-invertivores | Large invertebrates | *Balistapus undulatus* |
| Micro-invertivores | Small invertebrates | *Coris batuensis* |
| Pisci-invertivores | Fish and invertebrates | *Lethrinus olivaceus* |
| Piscivores | Fish | *Epinephelus merra* |
| Planktivores | Plankton | *Abudefduf sexfasciatus* |
| Spongivores | Sea sponges | *Pomacanthus sexstriatus* |

## STATISTICAL ANALYSES

All data analyses and figures were generated by using the Software R version 3.5.1. All the recorded fish species were included in the present analyses. Fish counts on each transect represented fish abundance. Therefore, the transect line was the statistical unit in our sample size. Overall, we ran two statistical models. We fit the first model to test for the overall change in total fish abundance before and after the disturbances. It was a General Linear Model, with a negative binomial distribution. The model had fish abundance as the response variable, while the period before and after the perturbation was fitted as a predictor with data collection site as a covariate. The model had the following structure: fish abundance ~ period of data collection + site. The model assumptions were checked with visual plots with the function influencePlot() in R language.

The second model tested for potential changes in the abundance within the 11 trophic-functional groups. Here, we fitted a zero-inflated negative binomial distribution due to the presence of many zeros in the count data. The zeros refer to the absence of some functional groups in the transects. The site identity was fitted as a covariate to control for potential differences between the two sites (R. Slobodeanu, 2018, personal communication). The model had the following function: fish abundance ~ functional group * period of data collection + site. As post hoc analyses for the second model, we ran least-squares means analyses with the function emmeans() from the package (emmeans in R language). The emmeans() function uses the Tukey method by default for multiple comparisons. The reported pseudo R-squared in the results are the Nagelkerke (Cragg and Uhler) values generated with the nagelkerke() from the package (rcompanion in R language) (see *Liu, Zheng & Shen, 2008*). For further details about statistical tests, R packages and script, please refer to our statistical script in the Figshare repository (DOI 10.6084/m9.figshare.4990919).

### Ethical note

The Animal Ethics Committee of the Queensland government (DAFF) approved the project (CA 2016/05/970 and CA 2017/05/1063).

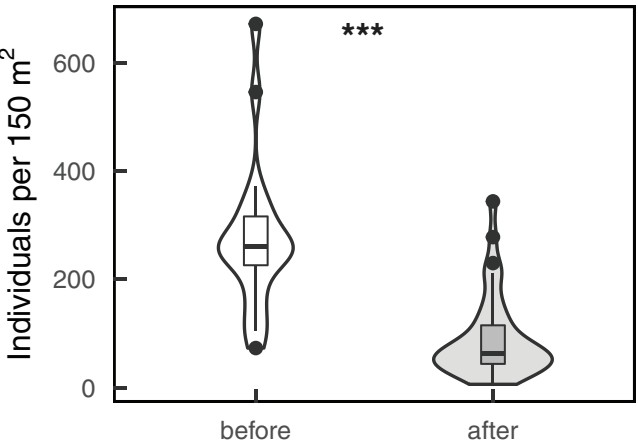

**Figure 2 Total fish abundance.** Boxplots are displaying median and interquartile of total fish abundance before ($n = 20$) and after ($n = 40$) the environmental disturbances (i.e., cyclones and coral bleaching). Negative binomial Generalized Linear Model: ***$p < 0.001$.

## RESULTS

Fish census data showed that total fish abundance significantly declined after the environmental disturbances (ANOVA: $N = 60$, estimate $= -1.239$, $X^2 = 52.885$, $p < 0.001$, pseudo R-squared $= 0.49$, Fig. 2), despite differences between the two study sites (ANOVA: $N = 60$, estimate $= 0.519$, $X^2 = 9.583$, $p = 0.002$). On the other hand, fish functional groups provided more details on where the decline in fish density occurred, with a significant interaction of the trophic-functional group and the period of data collection (ANOVA: $N = 660$, $X^2 = 68.899$, pseudo R-squared $= 0.66$, $p < 0.001$, Fig. 3). Post hoc tests showed that 10 out of the 11 functional groups went through a significant change in fish abundance after the disturbances, of which nine showed a decline (the contrast before–after): browsers (estimate $= 1.534$, $z = 3.427$, $p < 0.001$); corallivores (estimate $= 2.099$, $z = 2.418$, $p = 0.015$); detrivores (estimate $= 15.784$, $z = 3.610$, $p < 0.001$); excavator/scrapers (estimate $= 8.904$, $z = 4.308$, $p < 0.001$); grazers (estimate $= 11.836$, $z = 2.617$, $p = 0.009$); macro-invertivores (estimate $= 3.580$, $z = 2.195$, $p = 0.030$); pisci-invertivores (estimate $= 1.133$, $z = 2.337$, $p = 0.020$); planktivores (estimate $= 102.06$, $z = 4.340$, $p < 0.001$); and spongivores (estimate $= 39.951$, $z = 3.479$, $p < 0.001$). Only piscivores showed a significant increase in abundance (estimate $= -0.662$, $z = -2.277$, $p = 0.022$), while micro-invertivores were the only functional group that did not show any significant changes (estimate $= 6.956$, $z = 1.152$, $p = 0.249$).

## DISCUSSION

We identified a substantial decline in the density of reef fishes at Lizard Island following a sequence of severe tropical cyclones and coral bleaching. We documented a 68% decline in fish densities; a percentage close to what *Wilson et al. (2006)* found in their meta-analysis of 17 independent studies on fish density after environmental disturbances, in which an average decline of 62% was observed in fish density within 3 years after disturbances including cyclones and coral bleaching. These findings are in line with

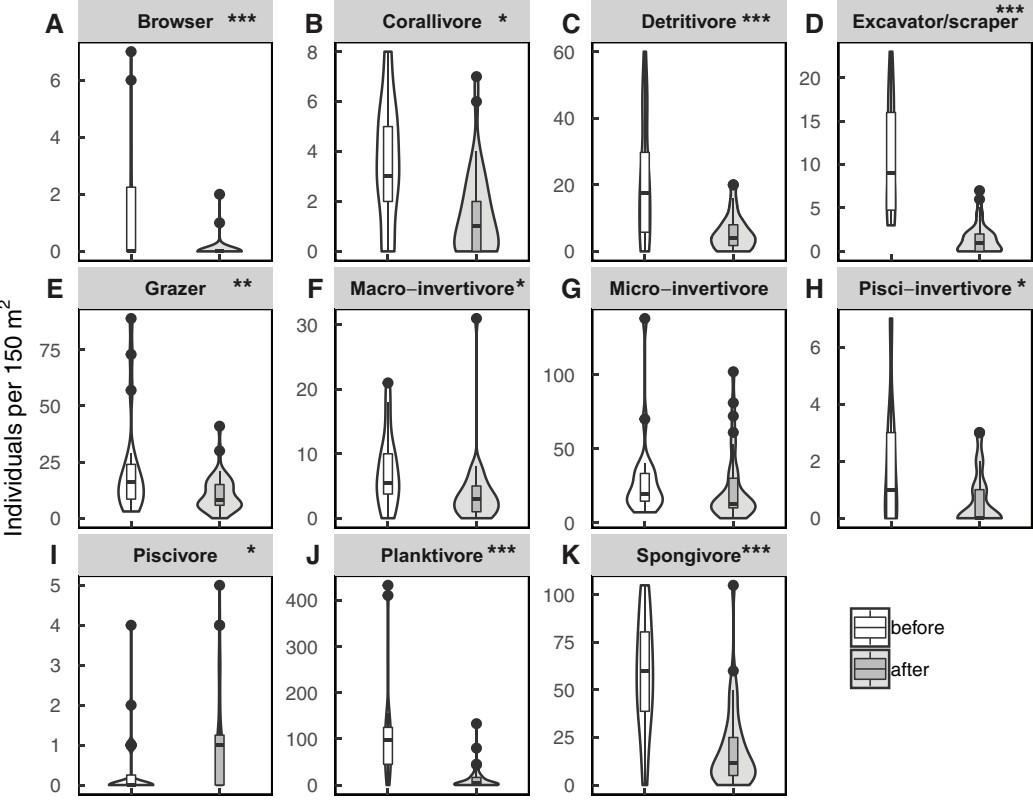

**Figure 3 Fish abundance per trophic-functional groups.** A to K are boxplots displaying median and interquartile of fish abundance within trophic-functional groups showing eventual changes from before ($n = 20$) to after ($n = 40$) the environmental disturbances (i.e., cyclones and coral bleaching). Note that due to the high variation in fish counts per functional group, the $y$-axes do not have the same scale. Post hoc analyses of a zero-inflated negative binomial model showing differences between before and after the perturbations within each functional group: $^*p < 0.05$; $^{**}p < 0.01$; $^{***}p < 0.001$.

previous studies suggesting that the loss of coral cover would lead to a reduction in fish density (*Jones et al., 2004*; *Russ & Leahy, 2017*; *Pratchett et al., 2018*). It suggests that the recorded decline in the present study might be due to the loss of coral cover. Cyclones usually destroy the reef structure, which would impede corals from possible rapid recovery (*Cheal et al., 2002*). Also, coral recovery might be compromised if the reef is repeatedly exposed to tropical cyclones over short-time intervals (*De'ath et al., 2012*; *Puotinen et al., 2016*). In addition to damage from cyclones, bleaching can reduce coral cover (*Diaz-Pulido & McCook, 2002*). Recently, *Stuart-Smith et al. (2018)* documented a 51% coral cover loss at the GBR after the 2016 bleaching event. Subsequently, it is expected that habitat loss would lead to a decline in fish abundance (*Pratchett et al., 2011*; *Brandl et al., 2016*). We acknowledge an important caveat in the present study: we were unable to incorporate information about the benthic habitat structure and benthic communities. This information would provide more insight into the fluctuations in reef-dependent fish communities (*Goren & Spanier, 1985*; *Holbrook, Schmitt & Stephens, 1997*; *Russ & McCook, 1999*; *Wismer, Hoey & Bellwood, 2009*; *Pizarro et al., 2017*; *Prazeres, Roberts & Pandolfi, 2017*; *Renfro & Chadwick, 2017*). Also, due to the absence

of data on the benthos, the significant effect of "site" in our model is virtually impossible to interpret (see Figs. S1 and S2). Nevertheless, the significant decline in fish densities recorded here can still inform us about the effect of extreme weather events on fish assemblages (*Wilson et al., 2006*).

Assessing fish density according to their trophic-functional groups showed a substantial decline in nine out of 11 functional groups. This suggests that most fish, regardless of their trophic affiliation, were susceptible to the disturbances. Nevertheless, the piscivore group was the only group to benefit from such disturbances. It is possible that due to the damage of reef structure and the resulting destruction of shelters, piscivores (i.e., reef-associated predators) would gain easy access to prey, from which they could benefit and thereby increase their numbers. Also, it is possible that bleached corals might no longer be suitable shelters for coral-dwelling species (*Coker, Pratchett & Munday, 2009*; *Pratchett et al., 2011*), nor appropriate camouflage background for small-bodied prey. As a consequence, predators would easily recognize their prey (*Phillips et al., 2017*), which would eventually change the assemblage structure of these predatory fishes (*Emslie, Cheal & Logan, 2017*). Nevertheless, such an increase might be transient in time and eventually be followed by a decline due to decreased numbers of prey. Also, the erosion of corals skeleton over time might result in a reduction of shelter and hunting options for ambush predators (*Kerry & Bellwood, 2012*).

*Graham et al. (2011)* predicted that micro-invertivores are one of the trophic-functional groups most vulnerable and macro-invertivores the least vulnerable to climate disturbances. Here, the micro-invertivores were the only group without apparent changes from pre- to post-disturbance. A potential explanation for this divergence is that micro-invertivores may show high functional redundancy, where losses in particular species can be replaced by population increases in other species that share a similar function (*Micheli & Halpern, 2005*; *Brandl et al., 2016*). Furthermore, we note that the decline in browsers, corallivores, and pisci-invertivores, as well as the increase in piscivores documented in this study, differ from previous results also collected around Lizard Island (*Ceccarelli, Emslie & Richards, 2016*; *Brandl et al., 2016*). One potential explanation is that those previous studies used post-disturbance data collected in early 2015, that is, only a few months after hurricane Ita hit the island, while we collected data 2–3 years after another cyclone and the El Niño event took place. A potential additional factor could be that the previous studies collected data in three and nine m depth, while our data include shallow areas of one to two m depth.

The trophic-functional groups that were most abundant pre-disturbance, the planktivores and spongivores, also showed a decline in numbers post-disturbances (Fig. 3). For instance, planktivores are mainly damselfish that are often highly coral-associated species (*Feary et al., 2007*; *Wilson et al., 2008a*), wherein habitat loss might explain the decline in their numbers. Such losses can be detrimental to the ecosystem balance, mainly because the planktivores play an important role in transferring nutrients from the pelagic environment onto the reef (*Pace et al., 1999*; *Fisher et al., 2015*). Spongivores also have a significant role in protecting corals by feeding on overgrowing sponges, thereby reducing coral-sponge competition (*Hill, 1998*). The decrease in fish density in the other

functional groups: browsers, detritivores, grazers, and excavators/scrapers, can also have severe consequences on the health and resilience of corals after disturbances. These trophic-functional groups feed on macro-algae, which prevent the latter from over-colonising the corals. Their functional role is hence beneficial for coral resilience, coral settlement, and growth (*Green & Bellwood, 2009*; *Cheal et al., 2010*; *Rasher, Hoey & Hay, 2013*).

## CONCLUSION

Environmental disturbances are expected to increase in frequency and magnitude due to global warming. Here, we found that such environmental events were followed by reductions in fish densities across multiple trophic-functional groups around Lizard Island. These findings add to the data that shows that future coral reef fish communities are susceptible to significant changes on this island. Supported by the larger scale fish assemblage changes across the GBR shown by *Hughes et al. (2018)*. It suggests that such losses can impact the functionality and stability of these communities (*Green & Bellwood, 2009*; *Rasher, Hoey & Hay, 2013*).

## ACKNOWLEDGEMENTS

We kindly thank the staff of Lizard Island Research Station for their support and friendship, R. Slobodeanu for his assistance with the statistical analyses, and J. McClung and Y. Emery for their contribution with proofreading.

### Funding

Funding was provided by the Swiss National Science Foundation (grant numbers: 31003A_153067/1 and 310030B_173334/1 to R.B.). The funders had no role in study design, data collection and analysis, decision to publish, or preparation of the manuscript.

### Grant Disclosures

The following grant information was disclosed by the authors:
Swiss National Science Foundation: 31003A_153067/1 and 310030B_173334/1.

### Competing Interests

The authors declare that they have no competing interests.

### Author Contributions

- Zegni Triki conceived and designed the experiments, performed the experiments, analyzed the data, contributed reagents/materials/analysis tools, prepared figures and/or tables, authored or reviewed drafts of the paper, approved the final draft.
- Redouan Bshary conceived and designed the experiments, performed the experiments, contributed reagents/materials/analysis tools, approved the final draft.

## Animal Ethics

The following information was supplied relating to ethical approvals (i.e., approving body and any reference numbers):

The Animal Ethics Committee of the Queensland government (DAFF) approved the project (CA 2016/05/970 and CA 2017/05/1063).

## Data Availability

Triki, Zegni; bshary, Redouan (2019): Fluctuations in coral reef fish densities after environmental disturbances on the northern Great Barrier Reef. figshare. Fileset. DOI 10.6084/m9.figshare.4990919.v1.

## Supplemental Information

Supplemental information for this article can be found online at http://dx.doi.org/10.7717/peerj.6720#supplemental-information.

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
