# Peer review of "Fluctuations in coral reef fish densities after environmental disturbances on the northern Great Barrier Reef"

_PeerJ, doi:10.7717/peerj.6720_

## Round 0.1 · original submission · Major Revisions

· Academic Editor

Major Revisions

This manuscript reports on valuable data, and an interesting question, but needs to be improved substantially. The reviewers highlight many issues and make many helpful suggestions on how it can be improved. In addition, I highlight the importance of getting the text right, in terms of clarity, grammar, formatting etc. Unlike most journals PeerJ does not have a copy-editing service, and although the reviewers highlight many places where the paper can be improved, ultimately it is up to the authors to ensure their paper is well written.

·

Basic reporting

Overall I was disappointed with this cursory treatment of a valuable dataset. I feel the authors have undersold their data, and could provide a much more in depth examination of the effects of cyclones and bleaching on reef fish assemblages. For example, while the authors state that Site 1 was more severely impacted by cyclones than Site 2, they provide no data to support this statement. At the very least, they should try and correlate changes in fish assemblages to changes in the benthos. Without such information, the authors are just documenting changes in fish abundance without trying to identify any causation. I also feel they have a chance for a more thorough examination of the effects of cumulative disturbances. I suggest they revisit the analyses and include some multivariate models, to look at how far the assemblage structure has shifted. It should be possible to partition the variation into effects due to cyclones and those due to bleaching, and draw some conclusions on which has the greatest effects. Its also interesting to me that the authors found such large reductions in abundance when a recent study, Ceccarelli et al 2016, found smaller changes or no effect for total abundance and family level abundance, while identifying substantial turnover and reductions in abundance of individual species. The authors should make a point of comparing and contrasting the studies, particularly as some of the sites overlap.
Much of the writing is laboured and difficult to read. The text should be improved to ensure that an international audience can clearly understand your manuscript. Some examples where the language could be improved include, but not restricted to, lines 38-40, 48-50, 54-55, 95-103, 180 to 181,255-257.
The introduction should be expanded to include the substantial literature that exists documenting changes to fish communities following disturbances. A more extensive appraisal of the literature would also allow the identification of knowledge gaps thereby providing a solid rationale to the study, which at the moment is lacking.

Experimental design

the research questions could be more well defined and articulated. I feel the analysis was somewhat cursory and could be much improved (see comments in Basic reporting).

Validity of the findings

Overall the paper presents valid, if somewhat cursory, results, although I would like to see the data re-analysed via a glmm rather than non-parametric tests. I would also like to see the analyses expanded to include diversity indices, an appraisal of shifts in assemblage structure via multivariate analyses, and to include information on changes to the benthos.

Additional comments

Specific comments

Abstract
Line 14 to 15 I’m not sure that I agree entirely with the opening sentence. There is considerable variation in modelling of future cyclone frequency and intensity, and three recent reviews concluded that the global frequency of cyclone is likely to remain stable or decrease by 2100 (Knutson et al., 2010; Christensen et al., 2013; Walsh et al., 2016), while another (Emmanuel 2013) predicted an increase in the mean frequency. However changes in the maximum cyclone intensity are more consistent with a tendency for increases in intensity and an increase in the frequency of the most intense cyclones (Emmanuel et al 2008, Knutson et al 2010, Christensen et al 2013).
Line 17 I don’t think evolved is the best term here. I would just ask how did they respond?
Line 22 to 24 this is ambiguous. I’m unsure as to whether the authors are stating that the site affected by only bleaching had greater decreases in fish density than the site hit by cyclones and bleaching.

Introduction
Line 36 Indo-Pacific
Line 39 replace ‘expulse’ with expel
Line 40 delete ‘fate’
Line 44 to 46 I don’t think that recovery can be compromised by only cyclones with short return times, the same is true of other disturbances and should be acknowledged here
Line 50 just state that it was a category 5 on the Australian scale
Line 69 to 71 – this is a very simplistic view of how reduction in complexity should affect predatory fishes. Not all predators are equal, and those that use ambush or stalking tactics may reasonably be expected not to fare well in areas where complexity has been reduced.
Methods
Line 82 – unless the transects were placed on the outside of North Reef, I don’t think you can claim that this site is exposed to the prevailing weather. From the text on line 81 it appears that the site was located inside Mermaid Cove.
Line 107 to 115 – how many observers collected the data? It appears that the 2011 data was part of another study (Wismer et al 2014). Was there any calibration undertaken among observers?
Line 117 – I would relabel as ‘fish trophic categorisation’
Line 126 to 127 – the use of Scarus sordidus is very outdated. It is now placed in the genus Chlorurus and the species name has most recently has been changed to spirulus in Australian waters, sordidus being the Red Sea/Indian Ocean variant
Line 142 – its usual to cite R rather than R-Studio.
Line 143 to 144 – I think the authors should reconsider their use of non-parametric statistics, and think about using generalized linear mixed models with either a poisson or negative binomial error distribution.
Results
Line 164 to 169 this could all be written succinctly in one or maybe two sentences.
Line 165 I would consider changing ‘Fish population size’ to something like total fish abundance or community abundance – population tends to refer to a population of a single species. Additionally, no uvc techniques will ever get an accurate estimate of a given population.
Line 179 to 180 this sentence is unnecessary – delete.
Line 180 to 181 – just say “there were significant decreases in the abundance of nine of eleven functional groups including, ….” There are numerous examples of this sort of poor writing throughout the whole manuscript. The manuscript would be vastly improved by a thorough edit aimed at improving the style and delivery of the messages. If English is not the author’s first language, I highly recommend that the manuscript is sent to a native English speaker for editing before re-submission.
Discussion
Line 195 to 197 – This sentence is awful, please re-write. I suggest something along the lines of “We identified large declines in the abundance of reef fishes at Lizard Island following a sequence of severe tropical cyclones and coral bleaching…”. This is really not a strong start to the discussion, which should deal with the most important findings of the study first.
Lines 199 to 200 – it is not clear what the authors mean by “the time lag between the perturbation and data collection appeared to be of major importance.” Can they please clarify?
Lines 205 to 206 – I think if the authors are going to discuss damage to corals, they should present data on change in hard coral cover.
Lines 206 to 208 the reference to coral recovery is odd because the time frame involved is far too short to be invoking recovery. More likely, it’s the reduction in live coral cover, and possibly habitat complexity which has caused declines in fish numbers. However, without any data on the benthos, its impossible to draw any meaningful conclusions about causality of the declines.
Line 209 to 210 I don’t think the authors can make this statement here, given they present no data relating to changes in habitat. In addition, there are many more relevant references than Pratchett et al 2011 that document the effects of habitat degradation on fish assemblages.
Line 211 to 213 I think this statement misses the point, and the authors should be discussing the compounding effects of successive disturbances.
Line 224 to 225 its hard to place this in context, as the authors present no data on changes in hard coral cover.
Line 255 replace reconstitute with constitute
Line 256 I think the importance of planktivores is overstated here. Their role is assumed to be important in bring pelagically derived nutrients onto the reef, however i think that how important this is in terms of reef functioning remains largely unknown.

Figures and Tables
Much of the information in Table 1 is redundant, given that abundance data is presented in Figs 2 and 3.

I wasn’t sure why the images of the sites were included in Fig 1. They don’t add anything to the story. Just the location map would suffice

Reviewer 2 ·

Basic reporting

Summary:
This paper looks at changes in the density of fish populations following disturbance from two hurricanes and a mass bleaching event on the North East GBR. It looks to show how different (dietary) functional groups respond to the storm and climate-induced mortality episodes at two sites with differing levels of impact.

1. The manuscript would really benefit from review from a native English speaker, or close copy-editing from the editor, as it feels as if this may have been written in a second language. The current phrasing, grammar order and flow of the writing makes it difficult to comprehend at times, and makes some sentences a little too loose. This occurs throughout the text (too numerous to list individually), but a couple of examples are shown below:
a. line 35: ‘A naturally occurring climate event is the El Nino cycle, it brings…’ would more sensibly be phrased ’The El Nino cycle is a naturally occurring climate event, it brings…’
b. line 40: ‘Henceforth, bleached corals face death fate it they do not re-establish…’ would make more sense as ‘Bleached corals die if they do not re-establish…’
c. The writers use the strange phrase ‘bleaching touched’… throughout which would be better written as ‘bleaching affected’ or something similar as bleaching is a process/ stress response.

2. The introduction would benefit from some re-working to give context and justification for the study:
a. The majority of text (line 33-57) focusses on the why bleaching and heavy storms are bad for corals, but then states on line 59 that the focus of this study is to provide data on fish abundances. So much more detail is needed initially on why these events might be bad for fish, and why we should be concerned if they are.
b. The final portion of the introduction (Line 59-71) says what the investigators did but doesn’t give a summary or any indication of what they are explicitly trying to show or test in this study.
c. There are a number of unjustified statements which if they were to be expanded upon a little would really improve the reader’s understanding of why analyses are being done. i.e. Line 62 – 63 states ‘To sort fish into functional groups, we opted for diet as the functional trait.’ Why did the investigators choose to look at functional traits, and why did they pick diet? Etc.

3. The article is written in professional structure, but there was no raw data shared.
a. Figure 1. I’m a little unsure why the work of Pizzaro et al. has been added to this given each DEM / orthomosaic only covers an area of around 10 x 10 m and only in 2015. I’m unsure what benefit this gives the reader as there were ten fish surveys conducted over 30 x 5 m at each site, each year. Locations of each survey and broad cover / slope if available would be far more useful. (Additionally, the figure terminology is incorrect here - image A and C are ortho-mosaics, whereas image B and D look to be Digital Elevation Models).

Experimental design

1. I liked the application of functional traits and thought the resulting changes following disturbance were interesting despite the limited geographical scope. However, I would have liked to see more discussion of why certain groups changed, and I think some of the conclusions didn’t quite match the observations. The main reason for this mismatch is that most of the introduction and conclusion state the importance of benthic composition and 3D structure for fish communities, however no assessments were done on the benthos, therefore much of this is speculation. It would be really good if benthic structure / composition / or even just broad cover could be incorporated in the analysis as it would give really valuable insight to the mechanisms of change.

2. The methods are somewhat unclear:
a. There is variability of survey depths. Were all surveys on reef flats? Please also clarify whether transects were parallel to the reef crest or to the shoreline (Line 109) as this is ambiguous and could lead to different communities occurring
b. Line 82: The authors state one of the sites is within an MPA. Does the other site allow fishing? This will clearly influence fish communities.
c. Line 87, 88 and 89 need references to show the described damages at each site following each disturbance.
d. Line 89: what data is this? Benthic cover or fish data? Could you summarise it?
e. Line 95-103: This paragraph has many repeated phrases and would benefit from restructuring. It is also unclear whether the fish census data was collected in the same way / same time of day etc. at each site and at each year. It would seem more sensible to place the paragraph 105-115 before this section to give clarity.
f. Lines 109-111: How did the divers calibrate in-water estimates of size to know what was smaller or larger than 10 cm? Also, it would be good to clarify why (and how) the surveyors measured the smaller fish over a 30 x 1 m area, but the big fish over a 30 x 5 m area.
g. Line 111: Why only record adult fish? Needs justification.
h. Line 112: Why separate surveys by 10 metres? Is this enough to have separate samples? Needs justification.

Validity of the findings

1. It seems likely that the sites ( particularly site 1) were already in an altered state before surveys, following the impacts of the 1998, 2002 and 2010 bleachings, which affected Lizard island, alongside the COTS outbreak (1995-1999), which dramatically reduced the cover in Mermaids cove (Site1). The community structure, dominated in 2011 by grazers, planktivores, spongivores, and detritivores would also indicate a very disturbed (non-coral dominated) community. Therefore it would be good if these large-scale disturbances were acknowledged and then discussed to give a clearer context and save conflating the effects of just recent hurricane damage and the most recent bleaching event.

2. While the discussion was interesting, much of the this and the conclusion is purely descriptive or speculative (primarily because of the lack of benthic data) - i.e. Line 234-237.
a. It is strange that it’s stated that there is an increase in piscivore density following disturbance in both the abstract and conclusion, given this increase is 0.6 to 0.73 per 100m2, and 0 to 0.46 per 100m2! Perhaps more note-worthy are the major declines in abundance in the spongivores, plantivores, detritivores etc. I think the highlighting of the piscivores in the abstract is a bit confusing / misleading.
b. Line 220-221: I’m not sure I understand this sentence, please could you clarify.
c. Line 226 -228: This sentence doesn’t logically flow. The writers state there is no statistically significant difference between 2014 and 2016, and then conclude an overall decline in fish numbers.
d. Line 228-231: This paragraph seems a bit superfluous to the study as brings in a whole new study that just looks at 2 individual species
e. Line 270-271: The authors state 'Our study highlighted the importance of continuous monitoring...' yet the data is (sporadically) annual.
e. Line 273-275: This conclusion is very speculative.

3. I thought it was perhaps a strange choice of statistical analysis to use non-parametric univariate tests. No justification was given for not applying a transformation for achieving normality and homogeneity.
a. Additionally, for this sort of community trait analysis I would suggest applying some multivariate tests such as a PERMANOVA would be useful to see how communities are changing.
b. Further clarity in lines 142 – 153 would be useful, i.e. whether fish abundance / density was averaged at each site etc.

·

Basic reporting

Overall:
The language and wording throughout needs a fair amount of improvement prior to publication. As is, some of the language it too informal or emotionally charged in places, too complex or convoluted and in some cases, is grammatically incorrect.

Abstract:
The abstract would benefit from more reporting of actual results and numbers throughout.

Introduction:
Topics need more in-depth introduction and room to breathe. Coral bleaching and zooxanthellae are discussed before corals or coral reefs are even introduced. Fish are arguably the main topic for the paper, yet there is no mention of them in the introduction until the aims paragraph.
There is also little to no reference to previous studies on the links between cyclones and bleaching, and fish abundances, or between coral and fish community structure in the main body of the introduction, however there are a number of these references in the discussion. I would suggest moving some of the newly introduced ideas and information from the discussion to the introduction.
The aim of the study is also unclear from the text. I can work out that the aim is along the lines of trying to detect changes in reef fish community structure in response to disturbances, but this needs clearly stating in text.
The final paragraph of the introduction itself also needs work. The final paragraph should state the knowledge gap/reason for the study, what the objectives of the study were and what the overall aim is. As is, links between fishes, and corals and disturbances are stated here for the first time, whereas by this point the introduction should have covered the background information and context for the study in sufficient detail to support the aims and objectives of the study, as well as what this study hopes to achieve in terms of broader context.

Methods:
The methods contain a lot of redundant/repeated information, and information on specific topics such as the study sites are found in multiple sections. This section should be clarified and written in a more structured manner, with all information for a specific component such as site, data collection, etc, all in the same section.

Discussion:
The discussion could be structured better, possibly by splitting paragraphs up by objectives. The conclusion paragraph also needs expansion and revision to set the results of this study back into the broader context.

Figures and tables:
Table 1 needs formatting for clarity prior to publication, and may benefit from being presented as a horizontal table rather than vertical.
Figure 1 may benefit from the inclusion of arrows on the map denoting where the two cyclones approached from to give better context to which site was more exposed.
Figure 2 needs some work as the line thickness around the boxplots is too high and the colour scheme makes viewing the 2017 data somewhat difficult, at least in the supplied pdf file.
Figure 3 has similar issues to figure 2 with some data very difficult to see clearly. It may also benefit from log10 transforming the y axis to help highlight variation at lower densities as well as allowing for the same y-axis to be used across plots, making cross group comparisons more intuitive.

Experimental design

Overall the design of the study is valid and the data tell a compelling story, however there is some ambiguity in places as well as issues in the analyses that, while not likely to invalidate the study, should be addressed prior to publication.

The two previous study datasets both mention “client species” for the fish surveys. Wismer et al doesn’t seem to provide a definition for this, nor does Triki et al, although Triki et al do mention that non-client species are removed prior to analysis. From the text it’s unclear whether you followed the same protocol of only recording client species, or recorded all species. Unfortunately, at time of review I’ve been unable to access the raw data to investigate this further. Clarification on this in text is needed.

Line 87: If you are going to state the damage as a fact you likely need references or data to back this up. Pizarro et al 2017 does touch on the cyclone damage somewhat, but doesn't state actual responses of sites to the cyclones. If this isn't possible I'd suggest contacting someone who knows how these sites responded to the cyclones and include a personal correspondence reference, or say that this site is highly likely to have been damaged by the cyclone and incorporate it as an assumption of the study.

I am somewhat unsure how the functional groups were assigned. Is it just diet? Or was diet assigned through a combination of traits? What scheme did you explicitly follow for classification? Two references are provided for the functional group classification but it’s unclear how they were implemented or integrated into a single scheme for this study. The paper might also benefit from a table with the group classifications, common names and species names, rather than a block of text.

The authors state that they consider the between years data as independent due to the time lapse between periods. I believe this is an incorrect assumption to make as the data for a given year seems at least partially contingent and the previous years’ data. Time and space are typically the biggest flags for non-independence, and while this may be ignored/assumed in some cases, it is not the case here. Independence is an assumption of the Kruskal-Wallis test, so this may preclude it’s use in this study. There are a number of other issues relating to the distribution of the data and the reporting of differences in means instead of medians as well. More details can be found at: http://influentialpoints.com/Training/Kruskal-Wallis_ANOVA_use_and_misuse.htm. Given this, I would suggest looking into a repeated measures approach that can account for non-independence. Also, the non-normality in the data may be alleviated with a transform such as log10 or square root, which may allow for the use of parametric statistics such as ANOVA.

At time of review I was also unable to access the raw data through the figshare link provided by the authors, and would need to inspect this prior to the paper being accepted.

Validity of the findings

Given the violation of independence, some of the results as stated are likely to change once a more suitable statistical analysis has been implemented. The reporting of significant differences in means throughout is also incorrect as the Kruskal-Wallace test compares differences in medians when the data are asymmetrically distributed as they are in this study. Reporting medians would also make the boxplots more intuitive for the reader as they present medians and not means.

However, from the figures and data some of the conclusions and results will more than likely hold up once a more suitable analysis has been undertaken. There are clear overall differences in overall fish densities between years, and there are also clear differences in some functional groups that will likely remain significant. However, some of the less clear changes in functional groups may or may not remain significant, but given the violation of independence, I would expect a few to be nudged outside of your selected critical value, and so should be discussed carefully.

That being said, there needs to be clarification in the methods as to how the 2011 and 2014 data were recorded and treated to ensure that classification and the targeted species are consistent across all datasets. As is, this ambiguity casts some doubt over the data used in this study.

I think the concluding paragraph is underselling the study somewhat. What you actually have here is evidence that disturbances predicted to increase in severity and frequency as climate change progresses will likely cause reductions in fish densities across multiple functional groups. The importance of this study is not really about the need for more monitoring, but is an example of how future reef fish communities may change across the GBR and wider coral reef range, with implications for long term coral health, economics and biodiversity.

Additional comments

Overall the results of this study are an interesting example of how reef fish communities respond to disturbances associated with climate change. The data highlight some clear differences in overall and functional group-specific fish densities between years, although the choice of statistical methodology to test for these differences is somewhat flawed.

The wording and structure of the paper also needs work to better tell a cohesive story with these results, setting the context for the study more fully, being more specific in the aims and objectives and discussing the results in a broader context. The language and writing are the biggest issues with the current manuscript, however they are able to be corrected with some work.

Given the data and potential of the results presented, I would suggest re-submission with major revisions. Particular care should be given to wording and structure to ensure that the text tells a cohesive and clear story around the results. The analysis should also be re-ran with a more suitable approach, however this will most likely not change the overall results but will increase their validity and perhaps de-emphasise some reported significance.

Additionally, I have made a number of annotations directly on the pdf to help guide where some changes and issues are using the following key;
Green star: wording/language issues
Blue star: unclear content/context, clarification needed
Red star: Error/inconsistency
Yellow star: Formatting issue
These annotations are not exhaustive, but mainly highlight key examples of issues mentioned in the review

---

## Round 0.2 · Major Revisions

· Academic Editor

Major Revisions

The reviewers provide excellent comments for how this manuscript can be improved. It is an important dataset that shows changes to the fish assemblage around Lizard Island. It adds to our understanding of the consequences of disturbances on coral reefs, but more can be inferred from the data than done with the current analysis. Two reviewers rightly point out that grouping the sites is an odd decision that weakens the paper. Perhaps there are good reasons to do so? But if so they should be spelled out. In fact, the paper still suffers from poorly defined aims.
Finally, the figures can be improved. Box plots show very little of the data. I recommend using raincloud plots, or at the very least bean plots.

·

Basic reporting

I was disappointed in the re-submission, as the authors have still presented a cursory treatment of the dataset, which should allow a much more thorough investigation in the effects of different disturbance types (cyclones vs bleaching) and sequential vs single disturbances (Mermaid vs Horseshoe). The glms used simply pool both sites together and look for changes before and after disturbance. This is flawed as the two sites have different disturbance histories and likely different fish and benthic communities as they are in different habitats on different sides of the island and pooling them together is inappropriate. The lack of benthic data is also a major flaw in this study. Although the writing has improved slightly, it still needs substantial work to bring it to an acceptable standard for publication in Peerj, where there is no copy editing.

Experimental design

The objectives of the study still need to be categorically stated. The analysis was cursory at best, and the authors need to consider whether the models used are appropriate for two sites that have different disturbance histories. There are also issues with non-independence of replicates and spatial auto-correlation

Validity of the findings

The findings are valid, ie fish declined in abundance after disturbances, but I question whether the models used are appropriate for reasons outlined above and in the specific comments

Additional comments

Specific comments

Abstract
Lines 15 to 16 – this is a weak statement. Global warming is predicted to increase the frequency and or severity of many disturbances including cyclones and bleaching
Lines 16 to 17 – This kind of emotive anthropomorphising happens a lot in scientific discourse, but I’m fairly sure Lizard Island hasn’t ‘suffered’. I suggest replacing ‘suffered’ with impacted by or something similar. Additionally, there have also been coral losses due to COTS activity. The statement ‘coral reef around Lizard Island’ is confusing. Do the authors mean the reefs around Lizard like Macgillvray, Nth Direction or simply the reef at Lizard Island. I’m sure the authors mean the latter but the way it’s written is ambiguous.
Lines 18 to 20 – this is a poor sentence. Fishes, not their density, are important component of coral reef ecosystems. They are generally measured using density but whether this is a good indicator of the impacts of disturbance is debateable. Measuring the impacts of a disturbance is usually done by examining the impacts to hard corals, and then the flow on effects of coral loss to reef associated fauna, like reef fishes. I suggest re-wording this sentence to something along the lines of “reef fishes are an important component of coral reef ecosystems and thus it is important to quantify the effects of disturbances on their abundance and distribution, particularly where disturbances severely degrade benthic assemblages which provide much of the habitat for fishes”.
Line 32 – see my previous statement concerning ‘suffering’

Introduction
Line 40 – I think the authors can be more emphatic here and state that extreme events ARE a great threat to coral reefs
Line 41 – not sure that ‘largest’ is correct. Consider replacing with most diverse
Lines 43 to 52 this section does not logically follow the preceding statement that losing live corals can have grave repercussions on the diversity and stability of this ecosystem. There’s too much focus on the mechanism of coral bleaching and virtually nothing on what it means to the abundance and diversity of reef associated fauna like fishes
Line 51 replace ‘ravage’ with something that’s not emotive. In fact the whole sentence could be re-written and state that cyclones can cause significant reductions to live corals, with flow on effects to reef fishes.
Line 54 – fishes as an indicator of habitat degradation is an outdated idea. Its much more robust to quantify changes in benthic communities to examine habitat degradation. Examination of changes to fish communities following disturbances is important in its own right, given the important role fishes play in reef trophodynamics and functional ecology.
Line 64 – insert ‘category’ before ‘5’
Line 66 – should probably also indicate what category cyclone Nathan was.
Line 66 – replace ‘earlier in 2016’ with ‘in the Austral summer of 2015/2016’
Line 66-67 just state that the GBR underwent extensive and severe coral bleaching across much of the central and northern GBR. The quoted 60% bleached coral cover is pretty meaningless
Line 68 replace ‘witnessed’ with ‘was impacted by’
Line 70-71 there is extensive literature showing the negative impacts of habitat degradation on the overall abundance of reef fishes and the abundance of different functional and taxonomic groups, and the authors should cite some of these.
Lines 73-76 this section is awkwardly stated and should be re-written.
Lines 76-78 the authors need to explicitly state the objectives of the study.
Eg the aims of this study were to document the effects of a sequence of disturbance on the fish assemblages at Lizard Island. Specifically, we wanted to: 1. quantify the change in the abundance of the total fish assemblage and that of different functional groups of fishes from before to after disturbances 2. Partition the changes in fish assemblages to individual disturbances events, [namely two cyclones and a severe bleaching event.
Or words to that effect. At the moment the objectives are still vague. In fact the whole last paragraph of the introduction needs a better structured rationale as to why the authors chose a functional approach – ie it gives insights into changes to important ecological functions performed by reef fishes, including herbivory which has been shown to be important in preventing turf and macroalgal proliferation following disturbance, thereby facilitating coral recovery
Methods
Line 94 replace ‘experimenter’ with ‘observer’
Line 106-109 this is vague. The surveys were not almost annually. Just state what months/years the fishes were surveyed. It is unnecessary to have ’the year’ before ‘2011’,’2016’,’2017’ (lines 107-8).
Lines 110-111 this is results. If not you need a reference
Line 133-134 the authors need to state that the model was run separately for each functional group if that is what they did. Furthermore, they need to explain what the fixed factors were in these models.
Lines 134-138 I’m concerned by the models. Given there was 3 years between the pre-disturbance surveys at Mermaid and Horseshoe and these two sites were differentially impacted by disturbances (Lines 110-111), I don’t think its valid to lump these together in the model. I suspect there should be main terms of year and site (and their interaction), with post-hoc analyses to test for different responses at individual sites through time. Alternatively, the authors could run separate models for each site, and indeed, this will be necessary if site is significant in the present model.
Furthermore, given that the transects were only 10m apart, I suspect that these samples are not independent as many fish species (eg parrotfishes, acanthurids, butterflyfishes, wrasses, snapper, trout etc) have home ranges much larger than this. Therefore, the authors need to account for non-independence and spatial autocorrelation. Using a generalised linear mixed model with transect as a random term will go some way to alleviating this issue.
Results
Despite now including more appropriate analyses, the authors still only have a cursory examination of the data, essentially lumping both sites together and examining changes in abundance before and after all disturbances. As the authors point out (Lines 110-114), these sites were differentially impacted by disturbances, Mermaid Cove was heavily impacted by both cyclones and bleaching whereas Horseshoe escaped the worst of the cyclones but affected by bleaching. Such differences in disturbance regimes provide an ideal opportunity to partition changes in fish assemblages among different disturbance regimes, particularly the effects of cyclones vs bleaching (which have different ramifications for habitat complexity through the removal of coral skeletons by cyclones but not bleaching), but also sequential disturbances (Mermaid) vs a single disturbance (Horseshoe). The results as present are at best a cursory examination, flawed by pooling sites with different disturbance histories. I am surprised the authors didn’t attempt to include any benthic data, which surely must exist given the number of researchers that visit Lizard Island. Instead they rely on anecdotal information.

Line 153-155 this sentence suggests that fish trophic group was fitted as a fixed factor, rather than the response variable. The authors need to clarify what the model structure was: ie response variables and fixed factors (see earlier comment in methods).
Lines157-163 this repetitive style of presenting statistics would be a lot more readable if the authors tabulated the stats
Discussion
Lines 176-177 well yes of course recovery will be compromised if multiple cyclones occur in a short time frame. The authors have an opportunity to examine the effects of sequential disturbances in this study (see previous comment in Results).
Line 178 replace the emotive term ‘devastating’ with ‘can reduce’ or ‘can impact’ or something similar.
Line 192-200 I think the authors need to caveat these statements to say that these increases were possibly short term only, and that as coral skeletons erode through time that there may be lagged effects on these fishes, particularly as there were declines in many prey species. As I pointed out in the previous submission, the loss of complexity can also be deleterious to ambush predators which utilise structure to hide and ambush prey.
Line 217 replace ‘evidenced’ with ‘underwent’
Line 219 planktivores are not the bottom of the food chain. Just say they play an important role in transferring nutrient from the pelagic environment onto the reef

Reviewer 2 ·

Basic reporting

I would suggest putting the supplementary figure 1 map as figure in the main text, as it is useful to get an idea of locations and distances between them etc.

• The raw data was unavailable to me. While it has a figshare DOI, I assume it is not publicly visible until publication?
• I think Figure S2 (B) is a little disingenuous as the x axis equal spacing and trendlines make it feel as though these intervals are even (while in reality they have gaps of 3, 2 and 1 years between datapoints. Similarly the y axis is on a strange (Log?) scale. I would suggest removing this figure section or put them both on normal continuous scalings.
• Table 1 – Please specify what size makes the invertivore diet ‘large’ or ‘small’

Experimental design

The hypothesis is that there will be changes in fish density post disturbance, but it would be good to say which groups we might see changes in and the reasoning for this, given that this is the main thrust of the paper. I.e. I would remove lines 216 – 227 of the discussion (which feels wrongly placed) and rework it into the introduction to let the reader know a little more why they should care if functional groups of fish were to change in this setting. I.e. expand the section lines 70-78.

The grouping of the two sites is a little strange given their differing disturbance histories. The Mermaids cove site has experienced cyclones and bleaching, while the other site has only experienced bleaching – therefore they have experienced very different types and magnitude of impact. I would suggest it would be more interesting to see the relative effects of cyclone + bleaching as opposed to just bleaching, or alternatively just make it explicit you are in effect just looking at bleaching effects (in the introduction and title) as it is the most recent, and you comment on the cyclone impacts in the discussion as you are not actually testing for a difference produced from the cyclone specifically.

There is no mention of whether assumptions of statistical tests (i.e. normality, homogeneity of variance etc) were met. Similarly the authors don’t actually state the test used to assess for differences (I assume ANOVA). Please can you add these in to the methods.

It would be good to be a little more precise in the writing here also, as it is currently ambiguous as to whether ‘site’ was a covariate for both tests, or just the second (line 138). Likewise in regard to post-hoc tests, were these just for the second model? (Line 139)

Validity of the findings

In light of the fact that by grouping the sites together the authors are only testing the effects of bleaching disturbance, in my opinion the title should more closely reflect this – rather than generalizing. The title should also probably specify the location or region (given the very small sample size) as the results cannot be generalized more widely than this island really. I.e. the title should be something like ‘Fluctuations in coral reef fish densities following bleaching disturbance on the Northern Great Barrier Reef’.

Similarly the abstract discusses cyclones as the disturbance but the paper does not tease apart these two impact variables.

Line 168 – we don’t know whether it was the bleaching or the cyclone (or something else) as the data is grouped.
I would remove lines 216 – 227 of the discussion as it is very broad and speculator
The conclusion needs to specify this study is only valid around Lizard island / the Northern GBR.
Line 232 should therefore read ‘…functional groups around Lizard island. This shows that…’
Line 233 – ‘…changes on this island, and potentially across the GBR. This study suggests that such losses can…’

Additional comments

Quite a few sentences would be more impactful if rephrased to state a fact rather than saying things may cause an effect. Other comments are just making the sentences more specific.

Abstract
This is a fact, so please change Line 15 – ‘Global warming does result….’
Change Line 17 – reef around Lizard island, part of the Great Barrier Reef, has recently…’
Change Line 19 – ‘is a good indicator to estimate the impact of such…’
Change Line 20 – ‘Lizard Island have had an impact…’
It isn’t in doubt that reefs are vulnerable to climate change so change Line 33 to 34 – ‘from the extreme weather events, leading to changes in the functional composition of the reef community’

Introduction
Please change the following lines:
Line 40 – ‘extreme events are a great threat…’
Line 43 – please remove ‘For instance,’
Line 45 – ‘El Nino event led to an increase in seawater…’
Line 50 – ‘bleaching, or any prolonged period of heating greater than 1 month...'
Line 59 – ‘Thus in the absence of fish, abundance…’
Line 70 – ‘Habitat degradation is known to have a negative impact on overall fish density, but exploring…’
Line 72 – ‘might be more informative as to the mechanism and effect of that impact. Bellwood….

Methods
Line 85 – Remove ‘More precisely, the study was carried out’ as unnecessary
Line 91 – I would make fig S1 just the actual Figure 1 as it is very useful context.
Please change ‘experimenter’ to surveyor throughout as this is an observational study. i.e. line 98.
Please leave a space between any number and the SI unit i.e. line 100 ‘ TL < 10 cm’
Line 107 – you say in a similar way – is this the same technique?
Line 152 – Remove ‘on the other hand,’

Discussion
Line 170 – please change to ‘…environmental perturbations, in which a decline of 62 % was observed in fish…’

·

Basic reporting

The language and wording throughout has been vastly improved throughout, although there remain a few wording and grammar issues that may be caught by a native English speaker on a second read through. I have also been able to access and examine the raw data and R scripts for the manuscript.

Overall there a few improvements to be made prior to publication, most notably, the aims and objectives in the final paragraph of the introduction are still not as clear as I would hope for.

Abstract:

Overall the abstract conveys the mains points of the paper well.

(Line) 15 - “Long-lasting environmental peturbations” doesn’t make sense as perturbations are typically thought of as a short-term phenomenon. I would also suggest swapping perturbations with disturbances throughout the text for clarity in most cases.

27 – “lost fish” is an odd word choice, I would use “reduced in abundance” or something similar.

32 – drop the “some” to make the wording more direct.

Introduction:

The introduction now follows a more logical structure than previously and includes necessary information (i.e. fishes). References to previous studies are more present throughout. The overall structure and flow of the introduction could be improved, for example the paragraph starting on line 54 begins about fish as indicators, but ends on line 68 about disturbances on lizard island. Broadly speaking, each paragraph should try to encapsulate a key idea, with each paragraph building on previously presented ideas and setting up for the next paragraphs.

The overall aims and objectives are still unclear and need more explicitly stating in the text. A paragraph highlighting the gap/need first e.g. fish are reliant on habitat provided by corals, which are under threat from disturbances. Understanding how fish communities react to… etc. Then state the aim of the study explicitly i.e. “Therefore, main aim of this study was…”. I’d then also follow it up with smaller objectives/aims relating to the main steps and results that work toward the main aim, i.e. “to achieve this, we asked 1) how total fish abundances… etc. This should make it very clear to the reader what the study is about and to set them up for the methods and results.

Methods:

The methods are now much more logically structured and clear, although the information about disturbances on lizard island starting on line 110 should probably be included in the first paragraph about the study site.

Discussion:

The discussion has been improved from the previous submission, although the conclusion could do with being expanded on, as is it seems a touch abrupt and undeveloped.

Figures and tables:

Table 1 is clear and is a useful addition to the manuscript.

Figure 1 is improved but could still do with some minor adjustments prior to publication. The figure legend should include what the perturbations (preferably disturbances) are, so the figure conveys enough information to convey the results alone. The significance stars (***) are also unnecessary as there’s only one comparison being shown here.

Figure 2 is also improved, but like figure 1, legend should include what the perturbations (preferably disturbances) are. It should also note that each plot has a unique y axis to make it clearer that between group abundances are different.

Experimental design

Overall the design of the study has been made clearer in the text and the new analysis is more appropriate given the data.

How the functional groups were assigned is now much clearer, and the included table with the group classifications, diet, and example species is useful.

Validity of the findings

With the more suitable analysis the validity of the results has been improved satisfactorily and are suitable for the aim of the study.

Additional comments

As with the previous version the results of this study is an interesting example of how reef fish communities respond to disturbances associated with climate change.

The paper is dramatically improved compared to the previous version, in terms of both clarity, statistical robustness and general structure.

There are a few minor issues and suggestions, however I'm happy to suggest minor revisions prior to publication.

Congratulations on the progress of the paper and the study overall.

---

## Round 0.3 · Minor Revisions

· Academic Editor

Minor Revisions

There are still many small edits required to ensure this manuscript meets standards of clarity for publication. These only constitute a minor revision, however I strongly encourage the authors to thoroughly check the manuscript for grammar and clarity and to address the other comments of the reviewers as well. Please note that PeerJ does not copy edit manuscripts as part of its production process, so it is up to the authors to ensure that the text is water tight.

Reviewer 2 ·

Basic reporting

Many thanks to the authors for addressing the issues relating to grammar and readability. The writing style is much improved and has clearly been worked on.

The structure, referencing and background appear sufficient.

There are still some typos and missing words, highlighted below:
• i.e. Line 34-35 should read ‘In summary, our findings provide evidence that the fish on the reefs around Lizard Island were considerably..’
• Line 76 should read ‘A suitable location to explore potential changes in fish abundance and functional…’
• Line 82 should read ‘we asked in to what extent fish communities would change…’
• Line 83-84 (this is pedantic, but): Pick either ‘post and prior’ or ‘before and after’, try not to mix terms.
• Line 91 should read ‘driven by a decline in fish species that rely directly…’
• Line 119 and 120, space needed between number and SI units, i.e. 5 m.
• Line 210 interpret misspelled

Experimental design

Line 59: The cited studies' effects rely on fish body size, which isn’t measured, so it is a little strange to use these studies to support the experiment (as this is not measured).

Line 60-62: These changes may provide some information on habitat loss, or it could just as well be a consequence of the direct abiotic factors you mention earlier. And the effect will be heavily dependent on whether the fish are directly reliant on live coral.

Line 64 -71. This paragraph needs work. The sentence gives the impression that all that is necessary to assess impact is the fish functional groups, rather than through measuring habitat/structure loss. I would say instead that in the Bellwood 2004 paper they argue that further insights can be gained from analysing fish functional groups, but only in addition to knowing the cause and extent of the disturbance events on habitat.

Line 68-69, the authors mention three functional groups being important but don’t specify which groups these are.

Line 70 -71 states ‘The functional role of these three herbivores is complementary and together their presence on the reef predetermine its resistance to disturbances (Bellwood et al., 2004)’ However The presence of certain herbivorous fish will only play some part in the resistance or resilience of a reef. Many other factors are important too. See Graham et al 2015 for example.
.
Line 83 – 87. I would suggest removing the sentence ‘To do so, we took advantage of fish survey data collected prior and after the disturbances within the frame of a study on the marine cleaning mutualism involving the cleaner wrasse Labroides dimidiatus (Wismer et al., 2014; Triki et al., 2018). The disturbances had a major impact on interaction patterns and cleaner wrasse strategic sophistication in laboratory experiments (Triki et al. 2018).’ These sentences don’t appear to have any bearing on the study.

Line 88-89: The authors state ‘Fish surveys were regularly collected at two sites (Fig. 1) that differ with respect to how they were impacted by the two cyclones’, but we really needed to know how these sites differ for this information to be useful.

Overall, I’m really struggling to see why the authors don’t just use linear mixed effects models here. I.e. the ultimate goal is to try to find if there is a difference caused by the disturbance events (and what that effect is) as well as seeing if there are differences in the extent of change between groups (and what those effects are), controlling for site differences. This equates to: Fish density ~ disturbance (Fixed effect; two levels) * functional group (interacting fixed effect; 11 levels) + (1|Site) (Random effect; two levels)
I think you would get a lot more robust, informative and clear results if you did this, as currently we really just know that there is a difference, not how much of a difference, the strength of the effects or how they may differ from different disturbance regimes.
An example is shown on line 172 -174. We can see by the statistic that the sites are different, but don’t know how they differ? Did only one site decline perhaps, and the other remain the same? (This uncertainty is stated by the authors in line 209-210). It would also be much more useful and informative to visualise this somehow, splitting the graphs by site / showing effect sizes.

While I think that the data on abundance change is interesting and the further splitting by function does give extra insight into how the community is changing, it feels like the study is still quite cursory given the species-level data available. For instance The authors could quite easily presumably assess fish community metrics such as species / functional diversity / richness / evenness etc , which I think would add to the interest of how the paper as a whole, and perhaps give some insight into how the community is actually changing rather than simply showing there are less fish.

Validity of the findings

Line 210 -213. I don’t think this can be stated, given you’ve just specified in the preceding paragraph that we don’t actually know what is driving the decline in fish abundance. As the authors don’t explicitly show the differences in abundance decline from the different disturbance regimes we don’t know if one has a worse effect than the other.

Furthermore, and I feel essentially to the whole paper is that we don’t know if this decline is just a generalised regional decline (i.e. due to some other factors like fishing / pollution / high temperatures affecting survival from changes in metabolism / feeding ability etc etc). It is unlikely to be, but we can’t say with certainty as there is no record of the extent of damage from the recent or historic disturbances. I would say that the paper’s premise is based on storms or bleaching affecting habitat quality, with a knock on impact for fish communities. However the authors don’t test for habitat quality / extent at any point (even broadly using remote sensing – where you could perhaps assess both cover and potentially structural complexity in shallow areas such as these sites). While the author states in the rebuttal letter that no such data was available, broad descriptive data could have been collected easily during survey as standard, and in line 236-238 they state ‘Furthermore, we note that the decline in browsers, corallivores and pisci-invertivores, as well as the increase piscivores documented in this study, differ from previous results also collected around Lizard Island (Ceccarelli, Emslie & Richards, 2016; Brandl et al., 2016).’. These cited studies each have both fish and benthic data from the same locations from roughly the same years, presumably giving the ability to at least broadly state changes in benthos.

I would further add that the lack of assessment of fish size structure change (and therefore biomass) makes the inferences of changes weaker. i.e. was it the smaller (more reef-dependent) fish that were lost primarily, will the loss of biomass be significant for future trophic dynamics etc ?

Line 221-228: This again is speculation as we don’t have any benthic data. With a similar problem with the statement on Line 194-196 (we don’t know if cover was lost, just that both areas bleached and one (?) had a cyclone - we can therefore only speculate).

Line 233 – 235 the authors state ‘A potential explanation for this divergence is that micro-invertivores may show high functional redundancy, where losses in particular species can be replaced by population increases in other species that share a similar function’. Could this not have been tested quite easily given species-level data was collected?

Line 236 – 238 The changes seem to be marginal i.e. corallivore median drops from 3 -1 over 150 m2, pisci-invertivore from 1 to 0, Pscivores up from about 0 to 1. I would feel uncomfortable reading too much into those changes (especially as they are averaged over two apparently quite different sites). There may well be different direction effects at the two sites. Furthermore, it seems strange to see so little change in corallivores (given the premise of the changes occurring is that bleaching and cyclones reduced coral abundance.

Line 244-250: Is there any indication as to what the authors think might be driving the planktivores and spongivore groups to declining abundances? These functional groups presumably aren’t directly affected by coral loss, unless through lack of habitat perhaps? It seems from Figure 2 it is only really these groups (and perhaps the detritivores and scrapers) that have had a strong change. Also, are the planktivores the damselfish? i.e. likely reduced in abundance from loss of habitat structure (as they are often highly coral-dependent), or is it a loss of food from seasonal or environmental changes / food abundances decline (over the three winter months of survey)?

Line 250 -255 These herbivorous fish would usually decrease from over-fishing rather than climate disturbance. You would expect that if there was a post disturbance shift to algal dominance these groups would actually go up in abundance due to increased algal dominance.

Line 260-262 Given that the data is quite speculatory, this uncertainty needs to be incorporated in the statements: i.e. ‘These findings add to the data that shows that future coral reef fish communities are susceptible to significant changes on this island. Supported by the larger scale fish assemblage changes across the GBR shown by Hughes et al 2018). It suggests that such losses might also impact the functionality and stability of the ecosystem.’

·

Basic reporting

Note: these comments are based on the supplied pdf not the tracked word document.

There are a few areas that need clarification and there are still a number of issues with wording that should be corrected prior to publication, including missing words. Some are listed here, but this is not an exhaustive list and the text again should be thoroughly checked prior to re-submission.

Abstract:

19 – missing word after “and a coral bleaching”

31:33 – On a second reading, this sentence is a bit unclear. In particular “affecting the stability of the ecosystem in itself”. Would the stability of the ecosystem be effected? Or would species compositions change to an alternative state (more piscivores, less corallivores, etc). It’s not clear what you mean by ecosystem stability here. I would perhaps stick to how fish assemblage composition is likely to change as this can be more easily inferred directly from your results.

34 – change “fish” to “fish assemblage” or something similar

35 – change “reef community” to “reef fish assemblage” or something similar. Although the reef community is likely to be affected by the disturbances, you only provide data for fish assemblages here.

Introduction:

42 – should be “Most diverse ecosystems”

43 – “Grave repercussions” should be replaced with simpler, more technical language.

46 – Make the link between climate change and prolonged el Nino’s explicit.

53 – It would be good to have a sentence at the end of this paragraph as it ends quite abruptly. The next section is about habitat degradation, so perhaps mentioning how both bleaching and cyclones change (/degrade) the environment might be useful.

55 – First sentence can probably be deleted.

57 – What aspect of fish? Their abundance?

73 – “changes in fish abundance and that in functional groups” – Awkward phrasing.

78 – “another severe cyclone, Cyclone Nathan category 4” – change to “Cyclone Nathan, a severe category 4 cyclone.” Or something similar.

79 – Remove/change “suffered”

80 – Might be worth noting that this was based on aerial and shallow water surveys, so excluded deeper reefs.

80 – “So for three years in a row, the island was impacted by a sequence of extreme weather events” – I’d put this earlier in the text and drop the “so for” as it’s a bit informal.

81 – “In the present study” should be “In this study”, also might be good to start a new paragraph here.

82 – “How far” is an odd wording choice

83 – “Took advantage of” is odd as well. Instead say “used”

84 – “within the frame of a study on the marine cleaning mutualism” – is unclearly written

84:87 – Discussion of the cleaner wrasse papers shouldn’t be in the last paragraph of the introduction. The last paragraph should set up the main aims and objectives of the study. I’m also unsure how these results tie in with the rest of the paper.

87:88 – “Fish surveys…” this is methods.

89:90 – “Here, we compared overall fish densities…” this sentence is a bit clunky. “Here, we compare fish densities before and after disturbances both overall and by functional group” is clearer.

Methods:

100:101 – I would put the lat and long after “Australia”.

139 – Typo “abovementioned”

143:144 – Unsure what the date following the R software version is referring to? Also R is typically cited (e.g. “ using R (R Core Team 2018)” )

144:145 – “Fish abundance was represented by fish counts on each transect.”. Wasn’t the abundance converted to density (fish per 150m2)?

151 – Which model assumptions? How did you check them?

153 – No need for “on the other hand”

155 – Maybe explain why there were so many zeros (I’m guessing that some functional groups weren’t present in some transects?)

156 – “similarly with the first model”, odd wording.

Results:

My main comment on the results would be that it would be informative to include the coefficient estimates of abundance along with the significance tests. For example, corallivores (estimate = -2, z = 2.418, p = 0.015), grazer (estimate = -10, z = 2.617, p = 0.009). This allows the reader to see how much something has changed along with the significance scores in a way that is clearer than estimating from the boxplots.

Discussion:

189, 191 – “Decline in abundance”, “Decline in fish densities”. The authors should make it clear and consistent whether the results show abundance or density. As the data in the figures is presented as fish per 150m2 I assume it should be densities and not abundance.

211- I would remove “That emerges despite an effect of site”

212 – Change “robust negative consequences” to “effect”, also add “…on fish assemblages” at the end.

217:218 – “Turned out to be” needs rewording. “was” would work here.

219:224 – “Resulting destruction of shelters…” The fact that piscivores also increased at horseshoe following bleaching (first draft, figure 3) suggests that it might actually be more to do with the loss of live coral, rather than structure. And this is something you can discuss more directly if you examine site-wise differences.

245 – “also evidenced” odd wording

259 – It would be good to tie anthropogenic warming and its effect on disturbance regimes into the conclusion given that those are in the first sentences of both the abstract and introduction.

Experimental design

The authors have decided to stick with using site as a covariate rather than examining each site separately, and while the results of this choice are still valid, I agree with the other reviewers that making some attempt to explore between site differences would increase the depth of the paper.

Returning to the first manuscript figures (Fig. 2), there is an apparent difference between how the fish assemblage at mermaid cove and horseshoe reacted following the cyclones and bleaching. At mermaid cove, there was a large drop between the pre and post disturbance communities, but at horseshoe the drop only occurred one year following the bleaching. A possible explanation for the lag effect of bleaching could be that the loss of structure following bleaching is much slower compared to direct damage caused by cyclones. Without benthic composition data this interpretation is conjectural, but citing literature to support this and other possible interpretations, as well as stating that these are plausible, but unconfirmed, interpretations, would help.

Returning to figure 2 in the first draft also highlights a drawback in the current analysis which pools the 2016 and 2017 data as “after disturbance” that I missed in the second version. Pooling both years is masking a potentially interesting pattern in the data (e.g. lag effects), and so I would suggest labelling them as post disturbance, and 1-year post disturbance to explore these differences. Another example at mermaid cove is that abundance continues to drop one year after the disturbances, which is interesting and gives the authors more to discuss from this dataset.

Validity of the findings

The results and models haven’t changed from the previous version and are still valid.

Additional comments

As is, there are still a number of wording and language issues that need to be addressed prior to publication.

The results are still valid and worth publishing, however, it would be a shame to not explore the data and results more comprehensively, including making site-wise comparisons and keeping the 2016 and 2017 data separate to track potential lag effects.

As PeerJ does not base its decision to publish a manuscript based on novelty or impact, the current manuscript is suitable for publication after the suggested corrections have been incorporated. However, I would advise the authors to consider exploring the data more thoroughly to maximise the impact of the manuscript.

---

## Round 0.4 · accepted · Accept

· Academic Editor

Accept

Thank you for revising your manuscript again. I am satisfied that all issues raised in the review process have now been resolved. Congratulations!

#